# Evaluation of a Community Hospital-Based Residencies’ Intimate Partner Violence Education by a Domestic Violence Shelter Expert

**DOI:** 10.3390/ijerph20095685

**Published:** 2023-04-28

**Authors:** Veronica Takov, Ashley Harnden, Kegan Rummel, Mariah Burnell, Shannon McMann, Carmen E. Wargel, Corie Seelbach, James McQuiston, Grace D. Brannan

**Affiliations:** 1McLaren Macomb Hospital, Mt. Clemens, MI 48043, USA; 2Spectrum Health Gerber Memorial Multispecialty Clinic, Fremont, MI 49412, USA; 3Methodist Health System, Dallas, TX 75203, USA; 4Turning Point, Inc., Mt. Clemens, MI 48046, USA; 5GDB Research and Statistical Consulting, Athens, OH 45701, USA

**Keywords:** intimate partner violence, community hospital-based residency, domestic abuse shelter partner

## Abstract

Intimate partner violence, or IPV, is estimated to affect an estimated 10 million Americans. From 2015–2017 our community hospital-based residencies trained first-year residents to improve education in recognizing and screening for IPV. This retrospective cohort study’s goal was to analyze the longitudinal effectiveness of the educational program. The education was based on a curriculum created by Futures Without and the United States Office on Violence Against Women. The curriculum was taught by Turning Point, the local county provider for victims of domestic and sexual violence, and involved five hours of training. Physician Readiness to Manage Intimate Partner Violence Survey was used as the assessment tool. Residents were measured pre-, post immediate, and one-year post-education. Measures that include perceived knowledge and perceived preparation improved post immediate and one year after the training (*p* = 0.0001). Actual knowledge increased significantly post immediate but decreased after one year (*p* = 0.0001). The proportion of residents who screened patients and the proportion of patients who were screened increased post-intervention. The educational training provided by our local shelter improved residents’ performance in several of the categories tested, but most importantly, improved IPV practice post immediate and generally one year after.

## 1. Introduction

The World Health Organization defines intimate partner violence (IPV) as “behavior by an intimate partner or ex-partner that causes physical, sexual or psychological harm, including physical aggression, sexual coercion, psychological abuse, and controlling behaviors” [1]. IPV transcends all geographical and cultural boundaries and is found worldwide [2]. In the United States, IPV is considered a major public health concern, with uniform reporting guidelines at the state and national levels [3].

Victims of IPV include both sexes, although reports worldwide and in the US indicate females have a higher rate [1,4]. A Centers for Disease Control and Prevention report indicated that at some point in their life, “1 in 2 women and 1 in 4 men suffered from contact sexual violence, physical violence, and/or stalking victimization in the hands of an intimate partner” [4]. Worldwide, 1 in 3 women has suffered from IPV [1].

IPV may have a diverse impact on a victim’s health, including sexually transmitted diseases, depression, post-traumatic stress disorder, and many chronic illnesses [5]. Additionally, it has been linked to structural changes in the DNA of victims’ offspring [6].

Due to its common occurrence, there is a high likelihood that healthcare professionals, physicians especially, will encounter and treat those affected by IPV. Various studies have looked at screening rates across different healthcare settings. Screening rates by a healthcare professional in an emergency department or primary care clinic are estimated to be 10–12% [7].

Resident physicians in community hospital-based residency training programs are in a unique setting to identify, assist, and manage IPV victims. However, physician knowledge, level of comfort, screening, and skills for managing IPV patients are deficient [8]. A survey of medical students and primary care physicians revealed that half never talked about IPV to their patients, and there is a need to improve training [9].

In general, physician education regarding domestic violence recognition and intervention is lacking in many hospital systems. A recent scoping review found 57% of articles were mostly for medical students, with lecture and standardized parents as the common forms of training [10]. Most training sessions are delivered using lectures and standardized patients, and a standard curriculum is lacking [10]. Many of the current studies are in academic health centers or specialized clinics [10].

A few recent studies beyond the scoping review included a two-part, 2.5 h IPV training program for IM residents at an academic health center that improved overall knowledge, confidence, comfort, and preparedness when measured immediately after training [11]. One limitation mentioned was that they did not measure the training’s effect longitudinally. A multi-site study in Canada and the US focused on educating healthcare workers in seven academic fracture clinics found that a year after a 2 h training, significant improvements were attained [12,13]. The specialized clinics utilized a train-the-trainer model and an educational intervention built upon prior curricula developed by IPV researchers, surgeons, and psychologists initially created for orthopedic surgery trainees. In another study, a half-day training for orthopedic trainees in an academic setting included lectures on foundational IPV knowledge and recent research outcomes [14]. It found the training to be effective after 3 months. A study on Greek physicians and trainees indicated that results were not statistically significant one year after IPV training [15].

Our hospital embarked on improving domestic violence screening and recognition starting in 2015. We have partnered with Turning Point, a domestic violence shelter that provides outreach and helpful resources to victims. To our knowledge, we have not found a study that looked at the longitudinal effect of an IPV training program provided by a community domestic violence shelter expert on community hospital-based residency training programs. The primary objective of this retrospective study was to analyze first-year or Program Graduate Year One (PGY1) residents’ perceptions, practice, and knowledge regarding intimate partner violence before and after they complete the community domestic violence expert’s educational program.

## 2. Materials and Methods

### 2.1. Study Design

This is a retrospective cohort study. The study was approved by McLaren Health Care’s ethics committee (SARC 4.23.2019). All residents at this community-based hospital were included in the training sessions. Completely de-identified, matched pre- and post-data from Program Graduate Year 1 (PGY1) resident physicians’ participants were retrospectively accessed. A well-studied and established, validated tool, Physician Readiness to Manage Intimate Partner Violence Survey (PREMIS), was used to assess changes [16].

The PREMIS tool was administered pre-, post immediate, and one year post the IPV educational program implemented from July 2015 to July 2017. The study analyzed PGY1 residents’ knowledge and comfort level in discussing and addressing domestic violence with patients both before and after their educational intervention.

### 2.2. Population and Sample Size

A total of 20 matched cases were analyzed. Pre- and post-questionnaire responses of all participating residents at McLaren Macomb who completed training between 2015 and 2017 were included. This study employed convenience sampling of available complete pre- and post-data. Only de-identified, anonymized pre- and post-training data were accessed for evaluation. Due to the small sample size, we did not collect demographic other than gender or programmatic data to ensure the anonymity of the respondents.

### 2.3. Description of Educational Program and Assessment Tool

This curriculum was delivered and adapted for our community by one of the co-authors of this study Carmen E. Wargel, LMSW., a social worker at the Turning Point domestic violence shelter who has IPV expertise. Turning Point is McLaren Macomb Hospital’s county domestic and sexual violence provider. The training utilized the Project Connect curriculum called CUES, created by Futures Without Violence and the United States Office on Violence Against Women [17,18]. CUES stands for Confidentiality, Universal Education + Empowerment, and Support. It was unique in that it was solely delivered by our social worker IPV expert, and no physicians were part of the instruction. The training sessions delivered information about domestic and sexual violence, as well as specific best practices for confidentiality, universal education, referral, intervention, and harm reduction for patients. To our knowledge, there are no other studies utilizing this curriculum.

In the different sections of the training, participants worked on nine learning objectives across three main domains:

Recognizing and Responding to Domestic and Sexual Violence in a Healthcare Setting

How domestic and sexual violence impact the health of their patientsFour best practice interventions with patients on the topic of domestic and sexual violenceWhy healthcare providers should discuss domestic and sexual violence with their patients

Confidentiality, Universal Education, Handling Disclosures, and Supported Referrals

Sharing the limits of their confidentiality with patientsUniversal education on domestic and sexual violenceBest practices for handling disclosure of domestic and sexual violence from patientsA supported referral for patients experiencing domestic or sexual violence

Harm Reduction, Targeted Interventions, and a Survivor Speaker

Describe and demonstrate the use of harm reduction strategies with survivors of domestic and sexual violenceDescribe and demonstrate the use of targeted interventions with survivors of domestic and sexual violenceDescribe real-life interactions between survivors of domestic and sexual violence with healthcare providers

PGY1 residents completed an average of 5 h of training. The curriculum began with an introduction to the four evidence-based interventions (confidentiality, supported referrals, harm reduction, and targeted interventions). Educational materials were provided regarding the lethality of IPV, what to say to a survivor, harm reduction, and targeted interventions. One session addressed how to maintain confidentiality using best practices and how to make a survivor-centered mandated report. As an experienced educator on domestic and sexual violence, C.E.W. knew that information was vital but not sufficient to create change. Each skill was reinforced with role plays on evaluating an IPV survivor so that residents could develop their own language reflecting each concept and have an opportunity to practice it out loud. The training also featured videos and worksheet guides to reinforce the concepts of the previous lessons. A guest survivor speaker spoke about her personal experiences with many medical providers during and after being abused, providing valuable insight into a survivor’s perspective. During the final lessons, residents observed videos and learned the concept of targeted interventions and risks survivors face, including sexually transmitted infections, strangulation, and traumatic brain injury.

The evaluation utilized the PREMIS tool, a standardized, validated, and publicly available measure of physician knowledge and readiness and the educational program’s effectiveness in IPV training [16]. It is a 15-min survey that allows a comprehensive evaluation in four main areas: background knowledge, preparedness, opinions, and practice issues.

During the 3 year period, we adjusted the training schedule and the number of sessions. These changes proved helpful in maximizing resident attendance and attention. However, the content and practice stayed largely the same, relying heavily on a local survivor speaker and experienced educator, as well as Futures Without Violence videos, framing, and evidence.

### 2.4. Outcome Measures and Statistical Analyses

Outcomes were measured at three points: prior to the educational intervention (Pre), immediately after the intervention (Post immediate), and one year after the intervention (Post one year) using the PREMIS tool. Primary outcomes are the different domains of the PREMIS tool administered pre- and post-IPV education programs. Composite scores for Background, Actual knowledge, and Opinions defined by the PREMIS tool were generated and evaluated. In addition, individual item scores also were generated to provide feedback for continuous improvement. For Practice Issues, the composite score was not calculated due to missing data for several items. However, individual scores for those with enough data were generated.

Perceived preparation was measured by asking the residents their level of preparedness on a 7-point scale (1 = Not prepared; 2 = Minimally prepared; 3 = Slightly prepared; 4 = Moderately prepared; 5 = Fairly well prepared; 6 = Well prepared; 7 = Quite well prepared). Perceived knowledge was measured by asking residents how much they feel they know using a 7-point scale (1 = Nothing; 2 = Very Little; 3 = A little; 4 = A moderate amount; 5 = A fair amount; 6 = Quite a bit; 7 = Very Much). Opinions were measured by asking residents their responses to statements using a 7-point scale (1 = Strongly disagree; 3 = Disagree; 5 = Agree; 7 = Strongly agree).

Descriptive data such as percentages, frequencies, total and mean scores, and standard deviation were generated. A repeated measures ANOVA was performed on continuous variables to determine the presence of statistical differences. Pairwise comparisons with Bonferroni correction were performed on variables that were found to be significant. In the results tables, mean separation was represented by a letter, whereby means sharing the same letter are not statistically significant. Statistical significance was set at *p* < 0.05. IBM SPSS Statistics for Windows, Version 25.0 (Armonk, NY, USA) was used to analyze these data.

## 3. Results

Seven programs were represented during the training: Emergency Medicine, Family Medicine, Internal Medicine, Radiology, Orthopedic Surgery, Obstetrics and Gynecology (OB-GYN), and General Surgery. Of the participants, 61% were male.

Prior to their IPV training at McLaren Macomb Hospital, residents recorded an average of 1.39 h of IPV training. Immediately post-training, the average increased to 5.21 h, and one year later, had increased to 5.79 h (*p* = 0.0002).

Table 1 shows the composite scores for each of the main PREMIS tool categories. Under Background, Perceived preparation (*p* = 0.0001) and Perceived knowledge of IPV management (*p* = 0.0001) for post-mean scores were statistically significantly higher than pre-scores. While the Post one year mean scores were lower, these were not statistically significantly different than the post immediate scores. Actual knowledge of the IPV total score increased by a statistically significant amount post immediate; however, it decreased to a level Post one year statistically significantly lower than pre-intervention. Individual items for these subcategories will be discussed in detail.

Under the Opinions category, Preparation (*p* = 0.058), Workplace issues (*p* = 0.11), Alcohol and drugs (0.57), and Victim understanding (*p* = 0.21), mean scores were not statistically significantly different before and after the intervention. Means scores ranged between neither disagree nor agree (4) and agree (5). Since the composite scores for these subcategories were not statistically significant, we did not perform a per-item test for significance. However, it is important to note that Post immediate and Post one year mean scores were higher than Pre for both Preparation and Workplace issues but not for Alcohol/drugs. Legal requirements and Self-efficacy were statistically significantly higher post-intervention, although there is no statistically significant difference between Post immediate and Post one year.

### 3.1. Background

Table 2 shows the scores for the individual items for the Perceived Preparation subcategory. Scores for all the individual items reflect the trends of the composite score in Table 1. Overall, an increase in score was observed Post immediate, which then decreased slightly Post one year. Participants progressed from feeling ‘slightly’ or ‘moderately prepared’ to ‘fairly well’ or ‘well prepared’ immediately post-IPV training. The only exception to this trend was in conducting a safety assessment of the victim’s children, where the mean returned to the ‘moderately prepared’ region, a change that was statistically significant. Participants also believed they were less able to help an IPV victim create a safety plan than they were one year prior.

Table 3 shows the scores for the Perceived knowledge subcategory. Scores for all the individual items reflect the trends of the composite score in Table 1. Residents initially reported perceived knowledge prior to IPV training in the ‘very little’ to ‘moderate amount’ range. After formal IPV training, their perceived knowledge improved into the statistically significant range of ‘a fair amount’ to ‘quite a bit.’ One year later, their overall perceived knowledge persisted in a statistically significant trend in the ‘moderate amount’ to ‘quite a bit’ range. An exception, such as their perceived preparation, was a statistically significant decrease in developing a safety plan with an IPV victim from Post immediate to Post one year.

The percentage of correct responses for Actual knowledge of individual items is shown in Table 4. Of the 34 questions, about 70% (24) started with higher than 70% correct responses and stayed high. This indicated that although residents did not report a high number of IPV training hours pre-intervention, they were knowledgeable about the basics of IPV. The question about the strongest single risk factor for becoming a victim of intimate partner violence, which is being female, was a challenging question for participants. It started with a very low score, improved Post immediate but decreased again Post one year. The rest of the questions saw improvements Post immediate, but scores declined Post one year.

### 3.2. Opinions

Table 5 shows results for Legal requirements and Self-efficacy. Under Legal requirements, both Post immediate and Post one year training, residents significantly improved their awareness of legal requirements for reporting suspected IPV, child abuse, and elder abuse. Under Self-efficacy, immediately post-training, residents were better at asking new patients about abuse in their relationships. One year later, there was a slight decline in this response, though not statistically significant. Immediately post and one-year post training, residents agreed they increased their comfort level in discussing IPV with patients, were more able to detect IPV without asking a patient about it, and were better able to garner information to determine IPV as the reason for a patient’s illnesses. When it comes to matching therapeutic interventions to an IPV patient’s willingness to change, the gains post-training were lost after one year. No statistical significance due to the training was found with the residents’ ability to recognize victims of IPV by the way they behave.

### 3.3. Practice Issues

One practice issue question was centered on asking patients about the possibility of IPV when presenting with symptoms such as injuries, chronic pelvic pain, or eating disorders. The residents improved from a mean score of 1.52 (0.58) to 2.51 (0.79) Post immediate and 2.53 (0.52) Post one year (*p* = 0.0001).

Compared to a pre-intervention rate of 90% who did not identify IPV in the past 6 months, Post immediate and Post one year, this rate decreased to 60%. For those who identified IPV in the past 6 months, pre-intervention, 10% of respondents provided information to and counseled the patients. In addition to these actions, in both the post and Post one year intervention time periods, residents who identified IPV in the past 6 months also indicated counseling patients about options, conducting a safety assessment for the victim and the victim’s children, and helping the patient develop a safety plan. 

Pre-intervention, 79% were unsure, and 21% indicated there was a protocol for dealing with adult IPV at the clinic or practice, compared to post-intervention responses of 35% and 65%, respectively. Post one year of the intervention, the response was 41.2% and 41.1%, respectively, in addition to 17.7% who responded that there was no protocol in place.

On familiarity with institutional policies regarding screening and management of IPV victims, pre-intervention, 15% answered affirmatively. This number increased to 75% in both the post and Post one year periods.

Regarding whether they are in practice in a state where it is legally mandated to report IPV cases involving competent adults, the percentage of respondents who indicated Yes was 25% (Pre), 65% (Post immediate), and 40% (Post one year).

Figure 1 shows the percentage of new IPV diagnoses. Pre-intervention, only 6% of respondents indicated 1–5 new diagnoses, and more than 90% indicated no new IPV diagnosis. Post immediate and Post one year of the intervention, there were 60% and 65% new diagnoses, respectively, for both 1–5 and 6–10 new diagnoses, and the number of those reporting no new diagnoses decreased to 40% and 45%, respectively.

The number of residents who screened increased post-intervention (Figure 2). Post-intervention, training participants demonstrated screening of all new female patients and patients periodically and during an annual exam and those with abuse indicators.

Table 6 indicates an increase in mean responses post and one-year post. The exception is for hypertension which, while it did not significantly increase Post immediate, it increased significantly one year post, compared to pre-training.

## 4. Discussion

The IPV training’s biggest impact is on Perceived preparation, Perceived knowledge, Legal requirements, and Self-efficacy. In addition, increased new IPV diagnoses and the number of residents screened were also improved due to the training. Opinions on alcohol/drugs were not impacted, while the rest of the opinions increased but were not statistically significant. The residents’ performed well on the actual knowledge questions even before the training on about 70% of the questions. The challenge was in retaining performance on several questions Post one year. The heavy work burden of residents’ patients and other educational responsibilities could be a reason and points to the need for ongoing training in this area to reinforce the concepts.

A systems model approach is a more coordinated and comprehensive structure and process that link hospitals, clinics, physicians, and other healthcare providers with community resources such as hotlines, domestic violence shelters, and support group [19,20]. This approach has been recommended as one solution to the IPV problem [19,20]. Our partnership with our local domestic violence shelter, Turning Point, is one example of this model to increase knowledge, awareness, and action to alleviate IPV in our community. This partnership has also led to our hospital using the Futures Without Violence curriculum. Futures Without Violence is a nonprofit organization, and Project Connect was a federally funded initiative that involved a comprehensive approach from victim identification, linking to resources, and prevention [18]. Nationwide, Project Connect has impacted greater than 400,000 patients through training more than 7000 healthcare providers in over 80 clinical settings [18].

While most curricula found in a recent scoping review were taught to Family Medicine, Internal Medicine, and Emergency Medicine [10], we included Radiology, Orthopedic Surgery, OB-GYN, and General Surgery in our training. As a result of the broad range of practice settings residents in a community hospital-based residency program encounter, the training was applied to multiple real-life practice situations, which provided another layer of learning. For example, in the office setting, residents were trained to separate the patient suspected to be a victim of IPV from any other person in the room in a non-threatening manner, such as by asking them to provide a urine sample in the bathroom. Concomitantly, the literature would have previously been placed in the bathroom to allow the patient to signal to the office staff that they were in need of assistance. In an alternative setting example, a trauma victim can be provided the literature in a safe manner by providing them with a list of community resources in a manner that can be easily concealed from their abuser as they exit the hospital. In any setting, multiple factors may prohibit a patient from immediately leaving their abuser. Therefore, residents were trained to develop safety plans with the patient with the expectation of eventually delivering the patient to safety.

Several studies are of interest when comparing approaches and outcomes to our study. Similar to our study, the EDUCATE study evaluated the residents one year after the training [12,14]. It used a train-the-trainer model where one lead from each of the specialized fracture clinics was trained, who then, in turn, trained everyone at the site. The participants included not only orthopedic surgeons and surgical trainees but also nonphysician health care professionals and research and administrative staff. Our approach was focused on first-year residents in a multidisciplinary model. Similar to ours, they showed significant gains immediately post-training and one year after [12,13]. Their studies also showed the same trends, where immediately after training, there was a statistically significant increase in scores, but one year after, the scores decreased, albeit not statistically significantly, from the immediate post-training period. Both models point to the longevity of the gains from training but also indicate that refresher courses to bolster these gains against erosion could be beneficial.

In another study, training was delivered by two physicians who have expertise in IPV research with patients and curriculum development [11]. Their training was also delivered only to first-year Internal Medicine residents. They also used the experience of a local women’s shelter during training, but not to the extent conducted here. The direct insight and expertise of the Turning Point trainer provided our residents with up-to-date information and recommendations on how to interact with these victims and offer effective care.

The length of training delivery is another aspect worth noting. One study involved the delivery of a 45 min IPV training by a community IPV trainer from a domestic abuse referral source [21]. Participants involved family medicine physicians, residents, midwives, and nurse practitioners at multiple training sites of a community-based urban family medicine residency in Chicago [21]. However, the results were mixed. On the other end of the spectrum, in a study involving Greek General Practitioners (GP), a 9 h IPV training was delivered by a team comprised of a GP, nurse, psychologist, social worker, lawyer, and administrators from an abused shelter [15]. Results were compared to a control group with no training. The training did not result in a statistically significant improvement for residents.

As documented in the results, residents improved their overall number of hours in IPV training. One year after their training at McLaren Macomb Hospital, the total number of hours reported increased, suggesting they had received additional training. This additional training, while not documented as part of this study, likely occurred in the form of additional reading and/or formal training on the resident’s behalf.

While the training resulted in very positive results, certain areas of improvement were also uncovered. In both Perceived preparation and Perceived knowledge, residents showed a decreased overall comfort level in helping an IPV victim create and develop a safety plan. This suggests that further training specific to this aspect would be beneficial for the long-term retention of knowledge. Victim understanding showed insignificant change among the residents evaluated and is another area where training can be targeted in the future. Having the necessary skills to discuss the abuse with an IPV victim who is female, male, or a different cultural ethnicity continued to be an area of improvement after training.

Continuous improvement and education are also needed to increase and sustain a higher screening rate. One example is the Veterans Health Administration (VHA) experience. In 2014 IPV screening was rolled out in Veterans Affairs medical centers (VAMCs) [22]. Through this policy, over half of VAMCs had adopted IPV screening by Fiscal Year 2020, an exponential increase from one VAMC in Fiscal Year 2014 [22].

The training provided many practical aspects. Through a series of educational seminars, role-playing scenarios, survivor testimony, and education from crisis center social workers and more experienced physicians, residents learned to screen for, identify, and provide an appropriate response to a given situation. Many practical tips were discussed. The relatively small size and specialty diversity of the first-year residency class at McLaren was a great benefit. Having clinical faculty in the room and directly involved in conversations about the lessons also strengthened its importance. The other benefit is that while some specialties are more likely than others to embrace this topic, the multi-specialty participation meant that the needs of survivors across our hospital system would be met. In a larger hospital setting, a single specialty residency could replicate this model by emphasizing the importance of domestic and sexual violence by centering it in lectures, engaging teaching faculty, and modeling involvement from the highest levels.

The uniqueness of our project is partnering with our local domestic and sexual violence program to leverage their expertise and experiences with survivors in our community compared to a more usual training model where physicians may not have the day-to-day immersion in a survivor-serving setting. An area for future research and program improvement would be to pair a domestic and sexual violence advocate with a physician who is well-versed in this area. In this way, the physician could offer concrete examples from their own practice, could be a part of training other teaching faculty, and could model behaviors during clinical instruction on shifts.

Given the difficulty in managing IPV, pre-written scripts were provided to residents as a starting point for a conversation with the patient as well as to ensure no critical details were omitted. Focus is also provided on personal relationships to be developed as the patient exits a relationship involving IPV. When a referral to a community center is made, all local resources are provided, highlighting the name and office number of a social worker that the provider frequently works with so the patient can start a personal relationship with a social worker in the outpatient setting.

Other benefits of this training include improving the identification of IPV victims that may have otherwise been missed by insufficient provider training and increasing the comfort level of providers when dealing with emotionally fragile patients. This study provides information and a model for training future residents at community hospital-based residency programs.

There are limitations to this study. The small sample size, while beneficial for a more engaged discussion, is a limitation of the study from the analysis perspective. This precluded us from performing subgroup analysis. Future research should include a bigger sample size, potentially through a multi-center study. As this was a voluntary survey, missing values were unavoidable. In our setting, the attending physicians were not previously trained and therefore were unable to engage in the discussions fully. In a see one, do one, teach one model of learning in residency programs, this is a significant limitation to ongoing behavior integration. One recommendation is to provide training to attending physicians as part of the need for a more comprehensive approach to IPV education in residency programs. We also recommend further education and practicing communication skills with patients of different genders and cultural backgrounds. An example will be incorporating more small group sessions or workshops to have hands-on practice and role-playing to address these gaps.

## 5. Conclusions

The evidence-based educational training curriculum provided by our local domestic shelter improved residents’ performance in several of the categories tested but, most importantly, improved IPV practice Post immediate and generally one year after. Areas of improvement were also identified.

## Figures and Tables

**Figure 1 ijerph-20-05685-f001:**
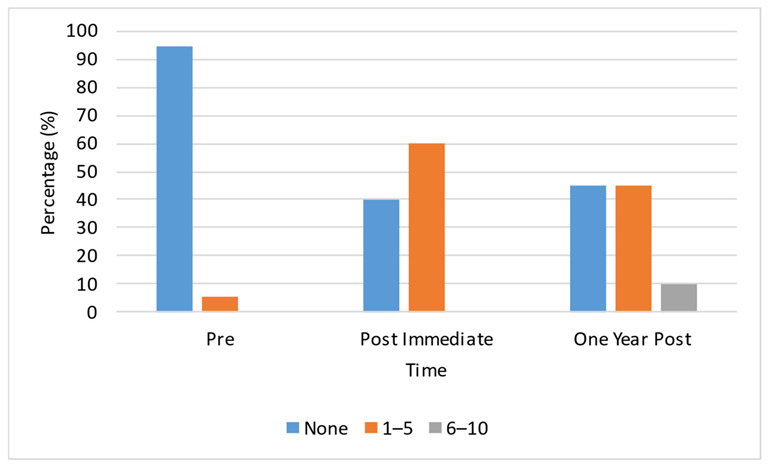
Percentage of new IPV diagnosis.

**Figure 2 ijerph-20-05685-f002:**
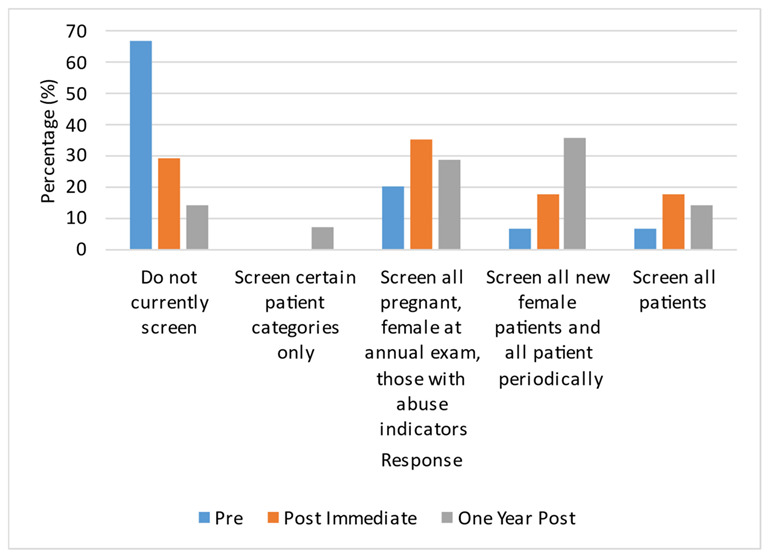
Percentage of residents who screened and which type of patients were screened.

**Table 1 ijerph-20-05685-t001:** Composite mean scores (standard deviation) for each of the main PREMIS tool categories measured at pre, post immediate, and post one year Intimate Partner Violence educational program training *.

Category	Time			*p*-Value
	Pre	Post Immediate	Post One Year	
**Background**				
Perceived preparation	3.6333 (1.18) b	5.6549 (0.76) a	5.4402 (0.63) a	0.0001
Perceived knowledge	3.52 (1.01) b	5.75 (0.76) a	5.37 (0.86) a	0.0001
**Actual knowledge**				
Total	26.70 (4.40) b	30.15 (2.80) a	22.90 (2.65) c	0.0001
**Opinions**				
Preparation	4.22 (0.98)	5.16 (1.1)	4.89 (1.09)	0.058
Legal Requirements	3.76 (1.24) b	5.32 (1.18) a	5.11 (1.32) a	0.0001
Workplace issues	4.02 (1.17)	4.69 (0.98)	4.53 (0.81)	0.11
Self-efficacy	3.42 (0.76) b	4.93 (1.11) a	4.68 (0.69) a	0.0001
Alcohol/drugs	4.33 (0.53)	4.3 (0.74)	4.18 (0.63)	0.57
Victim understanding	4.97 (0.45)	5.33 (0.94)	5.06 (0.95)	0.21

* For significant items, means sharing the same letters within each category are not statistically significant at *p* < 0.05.

**Table 2 ijerph-20-05685-t002:** Mean score (standard deviation) for perceived preparation questions measured at pre, post immediate, and post one year Intimate Partner Violence educational program training *.

Question	Time			*p*-Value
	Pre	Post Immediate	Post One Year	
a. Ask appropriate questions about IPV	4.11 (1.73) b	5.79 (0.79) a	5.58 (0.84) a	0.002
b. Appropriately respond to disclosures of abuse	4.37 (1.34) b	5.84 (0.76) a	5.47 (0.77) a	0.001
c. Identify IPV indicators based on patient history and physical examination	4.42 (1.57) b	5.95 (0.85) a	5.63 (0.5) a	0.0001
d. Assess an IPV victim’s readiness to change	4.11 (1.29) b	5.68 (0.75) a	5.53 (0.7) a	0.0001
e. Help an IPV victim assess his/her danger of lethality	3.78 (1.59) b	5.67 (0.97) a	5.22 (0.73) a	0.0001
f. Conduct a safety assessment for the victim’s children	3.63 (1.71) b	5.32 (1.11) a	4.95 (1.22) a	0.002
g. Help an IPV victim create a safety plan	3.32 (1.8) c	5.74 (0.99) a	5.21 (1.13) b	0.0001
h. Document IPV history and physical examination findings in the patient’s chart	3.83 (1.58) b	5.89 (0.96) a	5.83 (0.92) a	0.0001
i. Make appropriate referrals for IPV	3.06 (1.43) b	5.56 (1.15) a	5.67 (1.03) a	0.0001
j-1. Fulfill state reporting requirements for IPV	2.80 (1.67) b	5.50 (1.19) a	5.55 (1.1) a	0.0001
j-2. Fulfill state reporting requirements for Child abuse	3.05 (2.01) b	5.40 (1.19) a	5.50 (1.1) a	0.0001
j-3. Fulfill state reporting requirements for Elder abuse	3.05 (2.07) b	5.58 (1.22) a	5.47 (1.12) a	0.0001

* Scale (1 = Not prepared; 2 = Minimally prepared; 3 = Slightly prepared; 4 = Moderately prepared; 5 = Fairly well prepared; 6 = Well prepared; 7 = Quite well prepared). For significant items, means sharing the same letters within each category are not statistically significant at *p* < 0.05.

**Table 3 ijerph-20-05685-t003:** Mean score (standard deviation) for perceived knowledge questions measured at pre, post immediate, and post one year Intimate Partner Violence educational program training *.

Question	Time			*p*-Value
	Pre	Post Immediate	Post One Year	
How much do you feel you now know about: a. Your legal reporting requirements for IPV	2.95 (1.47) b	5.58 (0.69) a	5.26 (1.37) a	0.0001
How much do you feel you now know about a. Your legal reporting requirements for Child abuse	3.58 (1.87) b	5.79 (0.92) a	5.42 (1.35) a	0.0001
How much do you feel you now know about a. Your legal reporting requirements for Elder abuse	3.53 (1.84) b	5.84 (0.96) a	5.37 (1.46) a	0.0004
b. Signs or symptoms of IPV	4.10 (1.37) b	5.75 (0.79) a	5.50 (1.00) a	0.0001
c. How to document IPV in a patient’s chart	3.15 (1.09) b	5.70 a (0.8)	5.40 a (0.99)	0.0001
d. Referral sources for IPV victims	3.05 (1.18) b	5.74 (0.99) a	5.53 (1.17) a	0.0001
e. Perpetrators of IPV	3.16 (1.34) b	5.63 (1.07) a	5.21 (1.18) a	0.0001
f. Relationship between IPV and pregnancy	3.25 (1.45) b	5.65 (1.04) a	5.15 (1.09) a	0.0001
g. Recognizing the childhood effects of witnessing IPV	3.32 (1.67) b	5.37 (1.07) a	5.21 (1.08) a	0.0002
h. What questions to ask to identify IPV	3.79 (1.36) b	5.84 (1.01) a	5.63 (1.01) a	0.0001
i. Why a victim might not disclose IPV	4.05 (1.54) b	6.00 (1.03) a	5.55 (0.89) a	0.0001
j. Your role in detecting IPV	4.05 (1.51) b	6.16 (0.96) a	5.58 (0.9) a	0.0001
k. What to say and not say in IPV situations with a patient	3.85 (1.46) b	5.90 (0.97) a	5.55 (1.05) a	0.0001
l. Determining danger for a patient experiencing IPV	3.55 (1.39) b	5.85 (0.81) a	5.35 (0.99) a	0.0001
m. Developing a safety plan with an IPV victim	3.28 (1.49) c	5.67 (0.84) a	4.94 (0.87) b	0.0001
n. The stages an IPV victim experiences in understanding and changing his/her situation	3.53 (1.39) b	5.53 (0.96) a	5.37 (0.83) a	0.0001

* Scale (1 = Nothing; 2 = Very Little; 3 = A little; 4 = A moderate amount; 5 = A fair. amount; 6 = Quite a bit; 7 = Very Much); For significant items, it means sharing the same letters within each category are not statistically significant at *p* < 0.05.

**Table 4 ijerph-20-05685-t004:** Percentage of correct responses on Actual knowledge questions measured at pre, post immediate, and post one year Intimate Partner Violence educational pro-gram training *.

Questions	Time		
	Pre	Post Immediate	Post One Year
What is the strongest single risk factor for becoming a victim of intimate partner violence?	20	40.	25
Alcohol consumption is the greatest single predictor of the likelihood of IPV.	100	100	35
There are no good reasons not to leave an abusive relationship	100	100	30
Reasons for concern about IPV should not be included in a patient’s chart if s/he does not disclose the violence	100	100	55
When asking patients about IPV, physicians should use the words “abused” or “battered.”	100	100	90
Being supportive of a patient’s choice to remain in a violent relationship would condone the abuse.	100	100	65
Victims of IPV are able to make appropriate choices about how to handle their situation.	100	100	55
Healthcare providers should not pressure patients to acknowledge that they are living in an abusive relationship	100	100	90
Victims of IPV are at greater risk of injury when they leave the relationship.	100	100	65
Strangulation injuries are rare in cases of IPV.	100	100	80
Allowing partners or friends to be present during a patient’s history and physical exam ensures the safety of an IPV victim.	100	100	100
Which one of the following is generally true about batterers?	85	85	100
Which of the following are warning signs that a patient may have been abused by his/her partner—Chronic unexplained pain	90	100	95
Which of the following are warning signs that a patient may have been abused by his/her partner?—Anxiety	90	100	100
Which of the following are warning signs that a patient may have been abused by his/her partner?—Frequent injuries	100	100	100
Which of the following are warning signs that a patient may have been abused by his/her partner?—Depression	90	100	95
Which of the following are reasons an IPV victim may not be able to leave a violent relationship?—Fear of retribution	100	95	95
Which of the following are reasons an IPV victim may not be able to leave a violent relationship?—Financial dependence on the perpetrator	95	100	100
Which of the following are reasons an IPV victim may not be able to leave a violent relationship?—Religious beliefs	85	100	100
Which of the following are reasons an IPV victim may not be able to leave a violent relationship?—Children’s needs	95	100	100
Which of the following are reasons an IPV victim may not be able to leave a violent relationship?—Love for one’s partner	90	100	100
Which of the following are reasons an IPV victim may not be able to leave a violent relationship?—Isolation	90	100	95
Which of the following are the most appropriate ways to ask about IPV?—“Has your partner ever hurt or threatened you?”	90	85	80
Which of the following are the most appropriate ways to ask about IPV?—“Has your partner ever hit or hurt you?”	50	65	50
Which of the following is/are generally true?—There are common, non-injury presentations of abused patients	90	95	80
Which of the following is/are generally true?—There are behavioral patterns in couples that may indicate IPV	95	90	100
Which of the following is/are generally true?—Specific areas of the body are most often targeted in IPV cases	70	65	85
Which of the following is/are generally true?—There are common injury patterns associated with IPV	85	75	90
Which of the following is/are generally true?—Injuries in different stages of recovery may indicate abuse	90	95	80
Please label the following descriptions of the behaviors and feelings of patients with a history of IPV with the appropriate stage of change.—Begins making plans for leaving the abusive partner	89.5	89.5	100
Please label the following descriptions of the behaviors and feelings of patients with a history of IPV with the appropriate stage of change.—Denies there’s a problem	94.7	10	80
Please label the following descriptions of the behaviors and feelings of patients with a history of IPV with the appropriate stage of change.—Begins thinking the abuse is not their own fault	89.5	94.7	80
Please label the following descriptions of the behaviors and feelings of patients with a history of IPV with the appropriate stage of change.—Continues changing behaviors	68.4	94.7	85
Please label the following descriptions of the behaviors and feelings of patients with a history of IPV with the appropriate stage of change.—Obtains order(s) for protection	68.4	89.5	80

* Two questions were not included due to missing Pre or Post one year data.

**Table 5 ijerph-20-05685-t005:** Mean Scores for Legal Requirements and Self-Efficacy Subcategories of Opinions measured at pre, post immediate, and post one year Intimate Partner Violence educational program training *.

Questions	Pre	Post Immediate	Post One Year	*p*-Value
Legal Requirements				
I am aware of legal requirements in this state regarding reporting of suspected cases of: (a) IPV	3.42 (1.5) b	5.32 (1.34) a	5.11 (1.37) a	0.0001
I am aware of legal requirements in this state regarding reporting of suspected cases of: (b) child abuse	3.95 (1.75) b	5.37 (1.34) a	5.16 (1.34) a	0.002
I am aware of legal requirements in this state regarding reporting of suspected cases of: (c) elder abuse	3.89 (1.73) b	5.37 (1.34) a	5.21 (1.32) a	0.008
I comply with the Joint Commission standards that require assessment for IPV.	3.78 (1.11) b	5.22 (1.26) a	4.94 (1.51) a	0.006
**Self Efficacy**				
I ask all new patients about abuse in their relationships.	2.45 (1.28) b	4.05 (1.57) a	3.70 (1.26) a	0.0003
I am capable of identifying IPV without asking my patient about it. **	4.47 (1.07) b	3.47 (1.22) a	3.7 (1.08) a	0.005
I feel comfortable discussing IPV with my patients.	3.95 (1.27) b	5.58 (1.07) a	5.37 (1.01) a	0.0003
I am able to gather the necessary information to identify IPV as the underlying cause of patient illnesses (e.g., depression, migraines).	3.79 (1.13) b	5.21 (1.03) a	5.00 (1.00) a	0.0001
I can match therapeutic interventions to an IPV patient’s readiness to change.	4.11 (0.74) b	5.11 (1.33) a	4.63 (1.26) ab	0.025
I can recognize victims of IPV by the way they behave. **	4.11 (1.41)	3.53 (1.02)	3.79 (0.85)	0.176

* Scale: (1 = Strongly disagree; 3 = Disagree; 5 = Agree; 7 = Strongly agree). For significant items, means sharing the same letters within each category are not statistically significant at *p* < 0.05. ** Reverse-coded for reporting and analysis.

**Table 6 ijerph-20-05685-t006:** Mean responses to the question, “How often in the past six months have you asked about the possibility of IPV when seeing patients with the following condition?” *.

Item	Time			*p* Value
	Pre	Post Immediate	Post One Year	
a. Injuries	1.77 (0.73) b	3.46 (0.97) a	3.15 (0.80) a	0.0001
b. Chronic pelvic pain	1.20 (0.42) b	3.00 (1.41) a	2.70 (0.82) a	0.0001
c. Irritable bowel syndrome	1.33 (0.49) b	2.33 (1.07) a	2.08 (0.51) a	0.004
d. Headaches	1.25 (0.45) b	2.33 (1.07) a	2.25 (0.45) a	0.001
e. Depression/Anxiety	1.77 (0.83) b	2.92 (1.55) a	3.38 (1.04) a	0.0001
f. Hypertension	1.23 (0.44) b	1.38 (0.51) b	1.85 (0.69) a	0.032
g. Eating disorders	1.56 (0.73) b	2.78 (1.20) a	2.89 (1.27) a	0.003

* Scale (1 = Never; 2 = Seldom; 3 = Sometime; 4 = Nearly always; 5 = Always); For significant items, means sharing the same letters within each category are not statistically significant at *p* < 0.05.

## Data Availability

The data presented in this study are available on request from the corresponding author. The data are not publicly available due to privacy restrictions.

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
