# Peer review of "Evaluation of a Community Hospital-Based Residencies’ Intimate Partner Violence Education by a Domestic Violence Shelter Expert"

_ijerph, 2023, doi:10.3390/ijerph20095685_

Round 1
Reviewer 1 Report
The paper seeks to evaluate the impact of an evidence-based Intimate Partner Violence (IPV) training curriculum on residents' performance and IPV practice in a community hospital-based residency program. The authors have identified some noteworthy findings, such as improvements in perceived preparation, perceived knowledge, legal requirements, and self-efficacy. Additionally, the study indicates that an increased number of new IPV diagnoses and residents' screenings were achieved due to the training. However, the study has several limitations that should be addressed for a more robust and generalizable understanding of the topic.
The sample size is small, which, as the authors acknowledge, limits the power of the study and precludes subgroup analysis. This hampers the generalizability of the results to other settings and populations. To mitigate this limitation, the authors should consider expanding the sample size or collaborating with other institutions to create a multi-center study.
The study lacks a control group, which makes it difficult to determine if the observed improvements are attributable to the training or other factors. The authors should consider incorporating a control group in future research.
The authors mention that attending physicians were not trained in IPV, which could affect the residents' learning and practice. The authors should address this limitation by either providing training to attending physicians or acknowledging the need for a more comprehensive approach to IPV education in residency programs.
While the authors make efforts to compare their study with others, they do not thoroughly discuss the differences in the curriculum, training duration, or study design. A more in-depth analysis of how their study fits within the existing literature and why their training approach might be more effective would strengthen the paper.
The authors identify areas where the training did not lead to significant improvements, such as victim understanding and discussing abuse with victims of different genders and cultural backgrounds. However, they provide limited suggestions for how to address these shortcomings. More concrete recommendations for enhancing the training in these areas should be provided.
The authors mention the systems model approach as a potential solution to IPV, but they do not elaborate on how their study aligns with this approach or why it is relevant. A more detailed discussion of the systems model approach and its relevance to the authors' training and findings would enhance the paper's theoretical grounding.
Author Response
First of all, thank you to Reviewer 1 for your comments.
The paper seeks to evaluate the impact of an evidence-based Intimate Partner Violence (IPV) training curriculum on residents' performance and IPV practice in a community hospital-based residency program. The authors have identified some noteworthy findings, such as improvements in perceived preparation, perceived knowledge, legal requirements, and self-efficacy. Additionally, the study indicates that an increased number of new IPV diagnoses and residents' screenings were achieved due to the training. However, the study has several limitations that should be addressed for a more robust and generalizable understanding of the topic.
We agree, hence we have several recommendations for future studies throughout the paper.
- The sample size is small, which, as the authors acknowledge, limits the power of the study and precludes subgroup analysis. This hampers the generalizability of the results to other settings and populations. To mitigate this limitation, the authors should consider expanding the sample size or collaborating with other institutions to create a multi-center study.
We thank the reviewer for this comment. Due to the small sample size of our residency programs, we were not able to increase our sample size. However, one advantage of our pre- and post- test research design is the use of three responses per participant analyzed using repeated measures. Repeated measures have high statistical power because they allow the detection of within-person change over time (Guo Y, Logan HL, Glueck DH, Muller KE. Selecting a sample size for studies with repeated measures. BMC medical research methodology. 2013 Dec;13(1):1-8.). We do agree that future studies should strive for a bigger sample size. To this point, we included in our revision this sentence “This Future research should include a bigger sample size potentially through a multi-center study.”
- The study lacks a control group, which makes it difficult to determine if the observed improvements are attributable to the training or other factors. The authors should consider incorporating a control group in future research.
The study was not designed as a control vs treatment research. This was not the goal of the study. To achieve the goal of the study which is “to determine the longitudinal effectiveness of the educational program” we have to use a cohort approach where each participant serves as his/her own control over a time period of evaluation.
- The authors mention that attending physicians were not trained in IPV, which could affect the residents' learning and practice. The authors should address this limitation by either providing training to attending physicians or acknowledging the need for a more comprehensive approach to IPV education in residency programs.
We agree with the Reviewer hence it was part of the study’s limitation. We included the following sentence in our revision “One recommendation is to provide training to attending physicians as part of the need for a more comprehensive approach to IPV education in residency programs.”
- While the authors make efforts to compare their study with others, they do not thoroughly discuss the differences in the curriculum, training duration, or study design. A more in-depth analysis of how their study fits within the existing literature and why their training approach might be more effective would strengthen the paper.
We respectfully disagree with the Reviewer. We compared our study with many other studies. Among the multiple references we cited was a Scoping Review by Ghaith et al. (Ghaith, S.; Voleti, S.S.; Ginsberg, Z.; Marks, L.A.; Files, J.A.; Kling, J.M. A scoping review of published intimate partner violence curricula for medical trainees. J Womens Health 2022, 11, 1596-1613.) This review was a comprehensive look at IPV curricula to date.
- The authors identify areas where the training did not lead to significant improvements, such as victim understanding and discussing abuse with victims of different genders and cultural backgrounds. However, they provide limited suggestions for how to address these shortcomings. More concrete recommendations for enhancing the training in these areas should be provided.
We agree with the Reviewer. We added the following sentence “We also recommend further education and practicing communication skills with patients of different genders and cultural backgrounds. An example will be incorporating more small group sessions or workshops to have hands-on practice and role playing to address these gaps.”
- The authors mention the systems model approach as a potential solution to IPV, but they do not elaborate on how their study aligns with this approach or why it is relevant. A more detailed discussion of the systems model approach and its relevance to the authors' training and findings would enhance the paper's theoretical grounding.
Thank you for this comment. We revised/included the following sentences “A systems model approach is a more coordinated and comprehensive, structure and process that link hospitals, clinics and physicians and other healthcare providers with community resources such as hotlines, domestic violence shelters, and support group. This approach has been recommended as one solution to the IPV problem [19,20].”
Reviewer 2 Report
1. Lines 36 and 39: Please clarify the discrepancy between ten million out of 360 million affected by IPV and 33% of women and 25% having experienced IPV, seems like a lot more than ten million out of 360 million (or so).
2. Line 73 "trainees"?
3. Line 159 Either here or in the discussion please advise the reader on which adjustments seemed to help and why.
4. Line 170 Any useful patterns for the missing data to help other researchers avoid the same problems?
5. Table 1. Actual knowledge, total. How can the decline in total knowledge be explained? Were the answers too detailed to remember well after a year? Was the "knowledge" contradicted by actual experience?
6. Line 210 "ranged"
7. Line 253. Perhaps physicians who deal with case by case responses felt the risk factor was too stereotypical even if true; maybe they have seen cases of male victimization so often, they try to take each situation on its own merits.
8. Line 454 I do not think this is an appropriate response. I presume that informed consent was obtained at all testing times. If not, beware.
Author Response
First of all, thank you to Reviewer 2 for your comments.
- Lines 36 and 39: Please clarify the discrepancy between ten million out of 360 million affected by IPV and 33% of women and 25% having experienced IPV, seems like a lot more than ten million out of 360 million (or so).
Thank you for this insightful comment. This has been corrected in the revised paper by using the most recent CDC information available to avoid confusion.
- Line 73 "trainees"?
Corrected.
- Line 159 Either here or in the discussion please advise the reader on which adjustments seemed to help and why.
Addressed and clarified in the manuscript.
- Line 170 Any useful patterns for the missing data to help other researchers avoid the same problems?
Longitudinal evaluations are always tough. Increasing the sample size would minimize the effect of missing data. We included in our recommendations increasing sample size via a multi-site study.
- Table 1. Actual knowledge, total. How can the decline in total knowledge be explained? Were the answers too detailed to remember well after a year? Was the "knowledge" contradicted by actual experience?
The heavy work burden of residents' patient and other educational responsibilities could be a reason and points to the need of ongoing training in this area to reinforce the concepts. We added this to the manuscript.
- Line 210 "ranged"
Corrected
- Line 253. Perhaps physicians who deal with case by case responses felt the risk factor was too stereotypical even if true; maybe they have seen cases of male victimization so often, they try to take each situation on its own merits.
IPV is an overall complicated topic to deal with and explain. Hence more studies in the future are recommended.
- Line 454 I do not think this is an appropriate response. I presume that informed consent was obtained at all testing times. If not, beware.
We corrected it in the manuscript. It was waived by our IRB as it was a retrospective study.